# IoT-Enabled Classification of Echocardiogram Images for Cardiovascular Disease Risk Prediction with Pre-Trained Recurrent Convolutional Neural Networks

**DOI:** 10.3390/diagnostics13040775

**Published:** 2023-02-18

**Authors:** Chitra Balakrishnan, V. D. Ambeth Kumar

**Affiliations:** 1Panimalar Engineering College, Anna University, Chennai 600123, India; 2Computer Engineering, Mizoram University, Aizawl 796004, India

**Keywords:** cardiovascular disease, IoT, echocardiogram images, FCM, PRCNN, risk prediction

## Abstract

Cardiovascular diseases currently present a key health concern, contributing to an increase in death rates worldwide. In this phase of increasing mortality rates, healthcare represents a major field of research, and the knowledge acquired from this analysis of health information will assist in the early identification of disease. The retrieval of medical information is becoming increasingly important to make an early diagnosis and provide timely treatment. Medical image segmentation and classification is an emerging field of research in medical image processing. In this research, the data collected from an Internet of Things (IoT)-based device, the health records of patients, and echocardiogram images are considered. The images are pre-processed and segmented, and then further processed using deep learning techniques for classification as well as forecasting the risk of heart disease. Segmentation is attained via fuzzy C-means clustering (FCM) and classification using a pretrained recurrent neural network (PRCNN). Based on the findings, the proposed approach achieves 99.5% accuracy, which is higher than the current state-of-the-art techniques.

## 1. Introduction

Healthcare is a data-intensive process that runs and simultaneously produces new data every second [1]. It has been observed that advancements in system tools and software play an important role in the processing of healthcare datasets. This has brought healthcare and computing together to produce health informatics [2]. The mining tools work through an exploration of clinical data and the well-known experience of domain experts. The data mining process can act as an aid in finding rules leading to the creation and growth of epidemic diseases by utilising the extracted information from an immense quantity of data related to syndromes and healthcare records [3,4].

Data mining technologies for disease classification and prediction benefit healthcare providers by providing an application that helps detect disease at an earlier stage, rescuing patients, and reducing medical costs [5]. According to statistics from the World Health Organization (WHO), 12,000,000 deaths occur globally each year because of heart diseases. Globally, cardiac diseases are the primary cause of death and affect more people. Deaths from heart-related diseases are attributed to numerous pathological sequalae, including coronary artery disease (CAD), cardiomyopathy, cardiovascular disease (CVD), and so on, depending upon the circulation of blood throughout the body. CVD causes severe illness, disability, and death [6].

Reduced blood supply to the heart results from the occurrence of occluded coronary arteries, which leads to coronary heart disease (CHD), and subsequently results in myocardial infarction (MI), also known as a heart attack. This can also be due to blocking of the artery with plaques or fat deposits, which leads to the formation of blood clots. As a result, severe chest pain occurs as the heart muscle receives an inadequate volume of blood. In general, adults in developed countries such as the USA, UK, Canada, and Australia are more susceptible to heart diseases. Cardiac diseases can be defined as one or more complications of the heart [7].

The Internet of Things (IoT) has been conceptualised as a vast network of interconnected items. IoT is a cutting-edge technological paradigm that will have a big impact on future technology [8]. IoT has attracted considerable interest across a range of sectors and has also seen remarkable growth. Continuous connection and ubiquitous computing are not very difficult in the present networking era. At the beginning of the internet’s development, human and machine contact was constrained, but as internet technology advanced, communication between everything became possible, resulting in the Internet of Things (IoT) [9]. The most recent technology establishes a global network of machines and gadgets that are individually equipped with software that allows them to exchange and communicate information with one another through the internet. The important feature of IoT is that it can transform anything into an intelligent smart object by giving it the capacity to act, communicate, sense, and compute [10]. Figure 1 shows the performance of an IoT-enabled medical gadget.

The use of computers in today’s world has increased the ability to analyse and process pictures. A picture is stated as a variation in brightness from point to point. Images are digitised before applying any kind of processing technique. The process of converting discrete array points into brightness, grayscale shades, and grid points is known as digitization. An element in a digital picture is represented by pixels or point values [11].

Medical images have increased in number due to the advancement of imaging modalities, and the complicated nature of disease identification. Extraction of the biological region of interest from the background of a medical image is accomplished by segmentation [12]. The image will be segmented into different regions based on their relevance to pathologies, organs, and other biological structures during the segmentation process. Low-contrast values and the incidence of noise values are considered in the segmentation process of image processing. The automated process of segmentation plays a significant role in the analysis of images [13].

Segmentation’s accuracy determines the success or failure rate of a disease diagnosis system. Implementation of physical approaches and modern mathematical techniques, namely illumination, principal component analysis (PCA), convex property, partial differential equation, thresholding, and the significant material properties of images, has improved the accuracy of the process of segmentation. Robust segmentation techniques facilitate the assessment of diseases [14].

In terms of methods, neural networks present an interesting approach. Their use in classification and prediction processes due to its high potential. They are a famous and prominent tool for modelling data. A neural network is a self-adaptive and non-linear approach [15]. The relationship between the input and target patterns is discovered. A neural network is modelled after the brain’s learning behavior. It is used in training the complex data and the fields that generate complex data. A neural network has mapping abilities and can relate an input pattern to related output patterns. A neural network will learn with the assistance of examples and eventually be able to spot newly formed, untrained objects [16]. A neural network can predict new objects based on their previous development and has the ability to generalise the training. A neural network bears the capability to process the data in parallel, in a distributed manner, and at high speed. It is a fault-tolerant and strong processing system. In this paper, a deep learning framework for classifying heart disease and forecasting risk is developed.

One of the vibrant body parts in humans is the heart; it supplies each portion of the body by pumping or circulating blood to every portion of the body, including the brain [17]. If blood circulation to the brain and its diverse neurons in the nervous system is stopped in the heart, then the brain will die, which makes all tissue or nerves go to other parts of the body, where they stop working and lead a person to death. so that every person’s life eventually relies on the heart. Thus, the heart has a significant function in the human body [18].

The proper functioning of the heart assists the person in having a healthy and robust life. Forecasting cardiovascular disease is a more complicated and challenging task to accomplish through an instinctive analysis of the disease. Because a massive volume of data is warehoused in healthcare centers, analysing it is a difficult and complex task. Even though it is a difficult task, the identification of heart diseases is used in hospitals. It plays a prominent role in saving the individual’s lifestyle and making accurate and active decisions for beneficiaries [19,20,21]. The heart disease types and their general risk factors are given in Table 1.

The contribution of the work is to develop an early cardiac disease detection model using an IoT-module-based wearable sensor and deep learning classification techniques. Data from prehistoric heart patients and real-time data from IoT wearable sensors will be collected. If abnormality is detected, echocardiogram images will be acquired and the images will be pre-processed and segmented by using FCM (Fuzzy C-means algorithm).

To classify the echocardiogram images PRCNN (Pretrained Recurrent Convolution Neural Network) is used and risk prediction by using DCNN (Deep Convolution Neural Network) is involved in this work. The risk prediction is based on the classified results, and the patient can be treated accordingly.

The remainder of the research is structured as follows: Section 2 presents related works and a survey on heart-disease prediction are provided. The proposed classification and risk prediction methodology are illustrated in Section 3. The results acquired from the deep learning framework are detailed in Section 4, and concluding remarks are given in Section 5.

## 2. Related Work

Escamilla et al. (2020) [22] proposed a novel feature selection approach for extracting the significant attributes for classifying heart disease. The combination of Chi-square and principal component analysis (CHI-PCA) is utilised for feature reduction. The dataset used in this study was obtained from the UCI repository. The proposed feature selection with a random forest classifier achieved at 98.7% accuracy.

Ashir Javeed et al. (2020) [23] presented an intelligent prediction system using a floating window and neural network for feature selection and classification. The most commonly used Cleveland dataset shows that by removing the unwanted feature, the ANN and DNN approach performed the classification and predicted the heart disease effectively, obtaining 93% accuracy. This study attempts to achieve better performance using a deep learning classifier.

Khaled Mohamad Almustafa (2020) [24]: An attempt is made to properly classify the heart disease with a smaller number of attributes, which is more relevant for predicting the disease. A comparative analysis is performed by implementing the various classification approaches on the heart disease dataset. The unwanted attributes are removed using the Classifier Subset Evaluator. Among the other classifiers, the KNN achieved a great result of 99.70% and 0.69% RAE.

Devansh Shah et al. (2020) [25] designed a model in such a way that the discovery of intelligence from the clinical data is done automatically. Sometimes heart disease is diagnosed using patterns, as recurrent patterns are most applicable to the diagnosis of the disease. The learning algorithm is used to predict heart attacks based on significant weight and age and by using the designated patterns for effective prediction. The neural network learning algorithm is utilised to do effective analysis. Classifying heart disease was evaluated and concluded with justification of the importance of data mining in diagnosing and classifying heart disease. The dimensionality reduction of the features results in an increase in classification accuracy of 87.5%.

Anitha et al. (2019) [26] implemented a novel application in the healthcare industry in order to predict the disease at an earlier stage. One of the powerful data science tools is the R programming language, which is used in this study to perform the experiments with supervised ML techniques. The Naive Bayes approach performed great and produced 86.6% accuracy, whereas SVM and KNN obtained 77.7% and 76.67% accuracy in predicting heart disease, respectively.

The following Table 2 categorizes the study in different aspects with pros and cons.

Robinson Spencer et al. (2019) [27] described a machine-learning-based system for predicting heart disease. The experiment was performed by combining machine learning techniques with several feature selection techniques in order to select the significant attributes from the Cleveland data source. This study adopted several approaches, including PCA and Chi squared, for obtaining the relevant features. The classification model is then compared with other methodologies with possible error metrics such as RMSE, RAE, etc.

Latha and Jeeva (2019) [28] focused on improving the classifier’s accuracy through an ensemble approach. This mechanism enhances the accuracy of the weak classifier through bagging and boosting techniques. Through this approach, the classification accuracy can be improved by up to 7%. The bagging and boosting method was implemented on the heart disease dataset and achieved 85% accuracy.

From the literature, the identification of heart disease and the classification faces complexities in processing the image. The occurrence of errors in classification is a serious concern that reflects on the accuracy of the classification. Studies comparing the prehistoric medical data with real-time data collected from patients through IoT have not much been conducted in cardiovascular prediction. Echocardiogram images, which we consider in our research, are still challenging to classify robustly and accurately with the existing techniques. Hence, in this work it is famed with deep learning techniques for classification as well as prediction.
diagnostics-13-00775-t002_Table 2Table 2Issues in existing methods.S.NoAuthorDescriptionProsCons1Adem Atici et al., [29]Researchers used 2D spot-following echocardiography to detect severe coronary artery disease in patients with non-ST segment height myocardial localised necrosis. In total, 150 patients with NSTEMI who had experienced typical chest pain with unsound angina symptoms in the previous 24 hours were included in this study. Their cardiac capacities were assessed using 2D STE myocardial deformity analyses.Quick imaging. Easy to perform.Maintenance cost is higher.2Subhi J. Al’Aref et al., [30]Overview of the machine learning (ML) systems that are used to create predictive and inferential information-driven models. Here, a few ML applications in the fields of electrocardiography, echocardiography, and recently developed painless imaging modalities are highlighted including coronary course calcium scoring and coronary MR angiography.Better internal communication. Improvement in latency.Cost concerns are higher.
Limited Control and Flexibility.3Maryam Yahyaie et al., [31]The Internet of Things (IoT) facilitates online decision-production for anticipating a cardiac episode. In order to obtain current cardiovascular crisis information, an examination model was developed.Accurate results are gained. Deep analysis of data is performed.Minor errors in ECG signs are occurred.4Khushboo Bhagchandani et al., [32]Information analysis and the Internet of Things (IOT) can help reduce the delay in a number of ways, including addressing the patient’s situation, reaching dramatic guidance, or delivering the closest guidance possible. The suggested framework examines how sensors fit people who are predisposed to cardiovascular diseases and sends a warning to crisis contacts.A direct data transmission is possible. The information can be retrieved easily.There are chances that data breaching may occur.5Muhammad E.H. Chowdhury et al., [33]The momentum work suggests a wearable architecture for continuously identifying and alerting drivers to respiratory issues. The device, which consists of two interconnected subsystems that communicate wirelessly using Bluetooth technology, is highly accurate in distinguishing between ST-rise cardiac dead tissue and non-ST-height MI.Using a contactless ECG system
Faster in decision making.Accuracy of system is not considerable.


## 3. Methodology

In this section, the process of removing noise, filtering, highlighting the edges of the image, classifying the diseased image, and predicting the risk factors from the classified abnormal image of the heart by utilising a deep learning framework is detailed. The overview of this section is given in Figure 2.

### 3.1. Pre-Processing

When the input images are corrupted by noise, it becomes difficult to achieve better segmentation of medical images.Medical images are inevitably degraded by noise during acquisition, transmission, storage, and computation. The degradations in the medical images reduce the visual quality of the images and limit the precision and accuracy of image understanding and image analysis.

Gaussian noise is the most powerful noise, caused by the thermal agitation of electrons. As the amplifiers and detectors in the biomedical imaging instruments are the major contributing sources of Gaussian noise, it is also termed “electronic noise.” Gaussian noise disturbs each pixel of the entire image. The Gaussian probability distribution function models the Gaussian probability distribution of Gaussian noise corruption in an image, and the noise function n(g) with respect to the grey scale value “*g*” is expressed as
(1)$$n\left(g\right)=\frac{1}{\sigma \sqrt{2\pi }}\exp\left[-0.5{\left(\frac{g-\mu }{\sigma }\right)}^{2}\right]$$
where μ is the average value, which represents the peak in the Gaussian distribution curve, σ is the standard deviation.

Speckle noise, a multiplicative noise predominately occurs in ultrasonic medical imagining due to random variations in ultrasonic waves emerging from the ultrasonic scanner and insufficient application of electrolytic gel between the patient and the sensing head while capturing medical image. Speckle noise is a granular noise which degrades the medical image and reduces the image contrast in a great extent. The speckle noise follows gamma distributed probability function and it is mathematically defined as,
(2)$$n\left(g\right)=\frac{\left(g^{r-1}\right)\exp\left(\frac{-g}{a}\right)}{\left(\gamma -1!)\exp(\gamma \right)}$$

Low-pass filtering can be accomplished by an averaging filter or mean filter, and it permits only the frequency components that are below the cutoff frequency to pass. The averaging filter is specialised in removing spatial noise by replacing every pixel with the mean value of its neighbouring pixels, and this replacement is repeated for every pixel in the image. The low-pass filter removes the sharp contrast variations due to blurring and maintains the smooth region in the image. Here is an example of a simple averaging mask:(3)$$3\times 3\;averaging\;mask=\frac{1}{9}\left[\begin{array}{ccc} 1 & 1 & 1\\ 1 & 1 & 1\\ 1 & 1 & 1 \end{array}\right]$$

In general, an N×N averaging mask is given by,
(4)$$N\times N\;averaging\;mask=\frac{1}{N^{2}}\left(N\times N\;unit\;matrix\right)$$

In order to locate the centre pixel exactly, the feasible size of the mask can be odd. The blurring in average filtering affects feature localization. The averaging filter attenuates and disperses the impulse noise, but it is not completely removed. A Butterworth low-pass filter is used practically to mitigate these drawbacks.

The system function of the Butterworth low-pass filter is given by,
(5)$$H\left(m,n\right)=\frac{1}{1+{\left[\sqrt{\frac{m^{2}+n^{2}}{D_{0}}}\right]}^{2}}$$
where, *N* denotes order of the filter, $D_{0}$ represents cut off frequency.

Noise during acquisition and transmission affects digital images the most. An echocardiogram image’s quality is affected by parameters such as edge preservation, noise, and resolution. Thus, echocardiograms will inherently have noise, affecting the image quality for further processing, and causing problems in the proper diagnosis of a disease. Gaussian noise, salt and pepper noise, and speckle noise are among the types of noise that need to be considered and eliminated. Noise intensity can be described using the levels of noise density. There are five categories of noise level considered, namely 10%, 20%, 30%, 40%, and 50%, and based on the percentage, the echocardiogram images require denoising through the application of different filters.

The filters that can be applied for ultrasound images such as echocardiogram images include Median filter, Wiener filter, Kuan filter, Gaussian filter. Good denoise levels are indicative of an effective filter for removing noise and enhancing images. When this filter is used, the level of noise density is reduced and the values are given in Table 3.

As a result of the above observations, the Wiener filter appears to be effective in removing noise from echocardiogram images.

### 3.2. Segmentation

Fuzzy-based clustering algorithms, in general, do not require prior knowledge of training data and rely solely on the membership function, kernel function bandwidth, and clustering centre to enable unsupervised segmentation. The various tissues in a medical image possess overlapped grayscale intensities. Hence, fuzzy-based methods are highly preferred for the segmentation of medical images.

The most powerful and successful fuzzy-based clustering algorithm is fuzzy C-means clustering. Fuzzy C-means clustering (FCM) is a repetitive, unsupervised, iterative algorithm. Iterative optimization of the objective function is based on least square error. FCM partitions the image into a predefined number of C clusters and finds the cluster centre by minimising the dissimilarity function.
(6)$$J=\sum_{i=1}^{n}\sum_{j=1}^{c}U_{ij}^{m}{|x_{i}-v_{j}|}^{2}$$
where, *n* represents the number of one collection of sample set in a set of sample collection *X*.
$$The\;dataset\;X=\{x_{1},x_{2},\cdots ,x_{n}\}\in \;R$$
c−No.ofclusters;2≤c≤n−1
$$Cluster\;centres\;V=\{v_{1},v_{2},\cdots ,v_{c}\}$$

The weighting index, m∈[1,∞],m>1, the higher the value of m, the degree of fuzzy is greater.

Optimal segmentation is achieved by minimizing the dissimilarity function in Equation (Equation 6). Furthermore, Equation (Equation 6) clearly says that the Fuzzy C-Means partition is a c×n multidimensional matrix. The fuzzy membership matrix $\left(U_{ij}\right)$ and the clustering centers $\left(v_{i}\right)$ are updated by using the equations (Algorithm 1),
(7)$$U_{ij}={\left[\sum_{k=1}^{c}\frac{∥x_{j}-v_{j}∥}{∥x_{i}-v_{k}∥}\right]}^{\frac{2}{m-1}}$$
(8)$$v_{i}\frac{\sum_{i=1}^{n}\left(U_{ij}^{m}\right)x_{i}}{\sum_{i=1}^{n}\left(U_{ij}^{m}\right)}$$
fori=1,2,⋯,n,j=1,2,⋯,c

**Algorithm 1** FCM for Segmentation.
1.The fuzzy membership matrix $U_{ij}$ in Equation (Equation 7) is randomly initialised by satisfying the constraint.
(9)$$\sum_{i=1}^{c}U_{ij}=1,\;1\leq j\leq n,\;U_{ij}\in \left[0,1\right]$$2.Calculate centroids by using Equation (Equation 8).3.By using Equation (Equation 6), calculate the dissimilarity between data points and centroids and continue until its improvement over previous iteration is below a threshold.4.Calculate new fuzzy membership matrix $U_{ij}$ by using Equation (Equation 7) and proceed to step 2.


FCM iteratively updates the membership grades and cluster centers and routes the centers to the appropriate correct position within the dataset. As centroids are computed based on fuzzy partition matrix $U_{ij}$ which are initialized randomly, the final convergence of FCM does not ensure to an optimal solution.

### 3.3. Classification—Pretrained Recurrent Neural Network

Deep RNNs with gated recurrent units in the hidden layers with dropout symmetry information are briefly introduced in this section. Dropout is used on nonrecurrent connections to protect the state of any concealed units. To allow information to flow between time steps in RNNs, this is crucial. For the *l*th hidden layer of a multi-layered RNN with *L* layers, the hidden state $\left(h_{t}^{l}\right)$ is obtained from the previous hidden state $\left(h_{t-1}^{l}\right)$ and the hidden state $\left(h_{t}^{l-1}\right)$ of layer l−l. The hidden state transition from t−1 to *t* for layer *l* is given by function *f*:(10)$$f:h_{t}^{l-1},h_{t-1}^{l}\to h_{t}^{l}$$

The function *f* is implemented through the following transformations iteratively for t=1toT:(11)$$reset\;gate\;:\;r_{t}^{l}=\sigma (W_{r}^{l}\left[D\left(h_{t}^{l-1}\right),h_{t-1}^{l}\right]$$
(12)$$update\;gate\;:\;u_{t}^{l}=\sigma (W_{u}^{l}\left[D\left(h_{t}^{l-1}\right),h_{t-1}^{l}\right]$$
(13)$$proposed\;state\;:\;h_{t}^{-l}=\tan(W_{p}^{l}\left[D\left(h_{t}^{l-1}\right),\;r_{t}⊙h_{t-1}^{l}\right]$$
(14)$$hidden\;state\;:\;h_{t}^{l}=\left(1-U_{t}^{l}\right)⊙h_{t-1}^{l}+u_{t}^{l}⊙h_{t}^{-l}$$
where the product of Hadamard is indicated as ⊙, the vector of concatenation is indicated as [a,b] and D(.) is a dropout operator that randomly sets the dimensions of its argument to zero with probability equal to the dropout rate, $h_{t}^{0}$ the input $z_{t}$ at time *t*.

### 3.4. Risk Prediction—Deep Convolution Neural Network

The optimal features are classified with the assistance of a deep convolutional neural network with a rectified linear unit (ReLu) as an activation function. DCNN is composed of two stages, namely feature extraction and classification. The feature learning stage consists of a convolution and pooling layer. The fully connected and softmax layers are present in the classification phase. The deep CNN facilitates image feature learning, and the process of classification is simple. The outline of Deep CNN is given in Figure 3.

#### 3.4.1. Convolution Layer

Multiple filters in this layer slide over the input information, and the summation is done using an element-by-element multiplication method. The receptive rate of the input is then estimated as the output value of this layer. The weighted summation value is considered an input element for the subsequent layer. The focus area is slid to fill the other pixel values in the output of the convolutional layer. Every operation in the convolution layer is indicated by zero padding, stride, and filter size.

Rectified Linear Unit (ReLU) acts as an activation function and accelerates the convergence of the stochastic descent gradient. The implementation of ReLU is easy and exploited by thresholding, where the value of the activation function is mapped to zero. It returns zero if it receives a negative value, and t is returned if it receives a positive value. The RELU (AF) is given as,
(15)AF=max(0,t)

The gradient method stops learning when the AF value reaches zero, and the leaky ReLU is activated in that case. Its function is given as follows:(16)$${AF}_{l}=\left\{\begin{array}{cc} t & t>0\\ 0\times t & t\leq 0 \end{array}\right.$$
where the predefined parameter is indicated as o and assigned with the value 0.01.

#### 3.4.2. Pooling Layer

The pooling layer minimises the dimension of the output, and the most familiar max pooling technique is used to indicate the maximum pooling filter value. Max pooling is a promising approach that significantly reduces the input size. The maximum pooling technique outperforms summation and averaging.

#### 3.4.3. Fully Connected Layer

This layer learns the combination of non-linear information from the high-range features that is indicated by the convolutional layer’s output. The non-linear function in that space is learned by this layer.

#### 3.4.4. Softmax Layer

In this layer, the classification is performed, and the softmax function is exploited in the output layer, which is considered the normalised exponent value of the output information. This indicates the output probability and function are differentiated. Further, the exponential pixel value increases the probability to the maximum level. The soft maximum is equivalent to
(17)$${OP}_{x}\frac{\exp\left(z_{x}\right)}{\sum_{x=1}^{M}\exp\left(z_{x}\right)}$$
where the output of the softmax is indicated as opx for the output count x, zx is the output x before the softmax, and the total count of the output node is indicated as M. The class labels are categorised in this layer. The overall performance of the algorithm is given in Figure 3. The classification procedure is given in Algorithm 2.
**Algorithm 2** DCNN Procedure.
 Input ← Allprobablecombinationoffeatures
 TrainDCNN
 **for**
doinput=1toendofdatado
   **for** doneurons=1tondo
     **for** dorepeat=1tondo
      TrainDCNN
      DCNN−Storage←highesttestvalueisstored
     **end for**
   **end for**
   DCNN−Storage←effectivepredictionrate
 **end for**
 ReturnDCNN−Storage←effectivepredictionfromthecombinationofacquiredfeatures


## 4. Result and Discussion

### 4.1. Dataset Description

This section discusses the information about the dataset. The techniques used in Deep Learning assist in segmenting and classifying cardiac diseases. Dataset is derived from the open-source: https://archive.ics.uci.edu/ml/datasets/echocardiogram, (accessed on 27 January 2023). A total of 132 instances are included, 12 attributes are present, and the whole dataset is multivariate.

### 4.2. Experimental Results

The numerical outcomes of the proposed as well as existing techniques are presented in this section. The input image of echocardiograms is given in Figure 4, and the pre-processed image is given in Figure 5. The FCM based segmentation is accomplished in echo images that is illustrated in Figure 6.

### 4.3. Performance Metrics

The effectiveness and accuracy of the classification are investigated in terms of accuracy, sensitivity, and specificity. The proposed approach is compared with the existing state-of-the-art techniques, namely ANN, DNN [23], and SVM [26].

Accuracy: This prime performance measuring metric is defined as the ratio of true assessments to the total assessments. This image segmentation metric validates how well diseased cases are identified as positive cases and healthy controls are identified as normal people. The maximum performance of the segmentation algorithm is achieved when this ratio is at or near unity. This metric is mathematically given in Equation (Equation 18), and the results are given in Table 4.
(18)$$Accuracy=\frac{TP+TN}{TP+TN+FP+FN}$$

From the observation of the outcome in Figure 7, the proposed approach shows effective accuracy in the prediction of heart disease. The abnormal data is considered in prediction, and the DCNN is utilised for forecasting the risk of heart disease.

Sensitivity: Sensitivity is usually termed the “true positive rate”, which can be formulated as in Equation (Equation 19), and the results are given in Table 5.
(19)$$Sensitivity=\frac{TP}{TP+FN}$$

From the observation of the outcome in Figure 8, the proposed approach shows effective specificity in the prediction of heart disease. The abnormal data is considered in prediction, and the DCNN is utilised for forecasting the risk of heart disease.

Specificity: Specificity is regularly called the “true negative rate”, which can be modelled by Equation (Equation 20), and the results are given in Table 6.
(20)$$Specificity=\frac{N}{TN+FP}$$

From the observation of the outcome in Figure 9, the proposed approach shows effective specificity in the prediction of heart disease. The abnormal data are considered in prediction, and the DCNN is utilised for forecasting the risk of heart disease.

A comparison is made between the proposed algorithm and various other algorithms such as ANN, DNN, and SVM. According to the results, the computation times for ANN, DNN, SVM and PRCNN are 46 ms, 29 ms, 32 ms, 17 ms, respectively. So, the calculated result is more efficient for our proposed algorithm and has less running time than that of others.

## 5. Conclusions

Cardiovascular disease currently represents a key health issue and is largely responsible for the rising death rate. Due to the rising death rate, healthcare is an important area of study, and the knowledge gained from this health data analysis can help in the early detection of illness. Retrieving medical data is becoming increasingly important for early diagnosis and prompt treatment. Segmentation and classification is a fascinating and thought-provoking area of research in the realm of medical image processing. A digital image is divided into a number of segments based on intensity, form, texture, or colour. The data gathered from Internet of Things (IoT)-based devices and patient health records are taken into account in this study. The preprocessed and segmented photos are further processed for classification and predicting heart disease risk using deep learning techniques. The segmentation is accomplished by FCM and classification by PRCNN. In this research, a pre-trained neural network is utilized for classification and the proposed approach attains a peak accuracy of 99.5%. The suggested method performs better than current state-of-the-art methods. In future, the approach can be extended with multi-class classification and with bio-inspired optimization approaches.

## Figures and Tables

**Figure 1 diagnostics-13-00775-f001:**
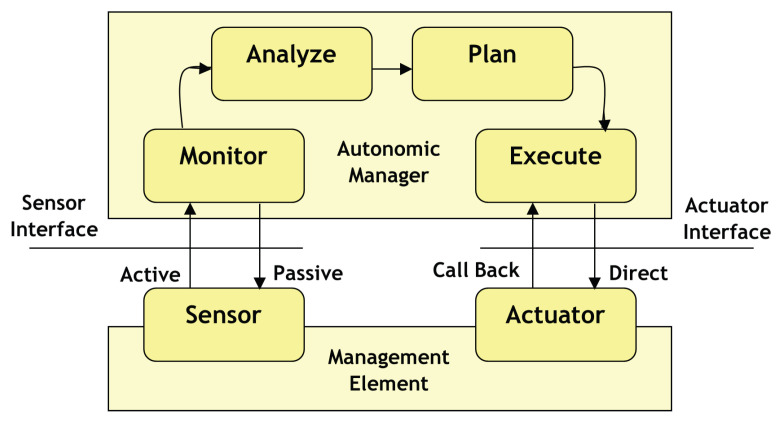
Sensor function in IoT enabled Medical Device.

**Figure 2 diagnostics-13-00775-f002:**
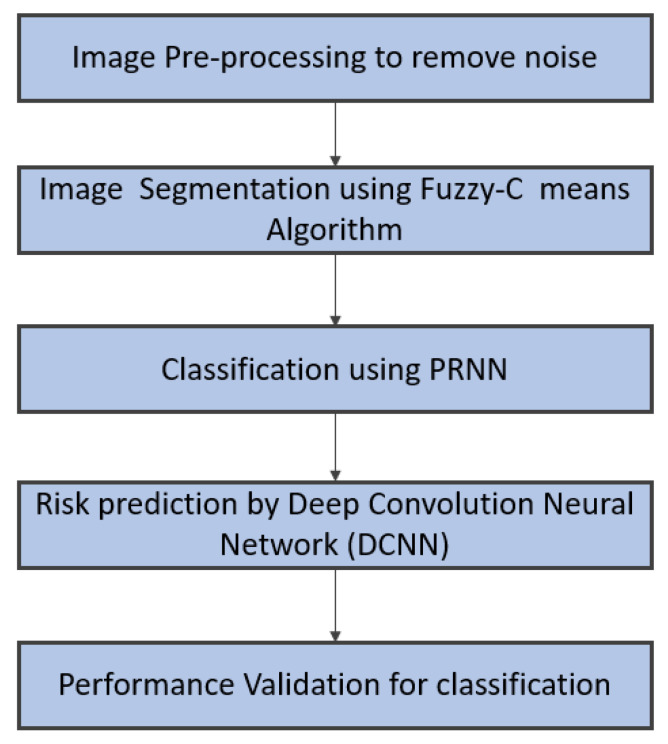
Overall workflow of proposed work.

**Figure 3 diagnostics-13-00775-f003:**
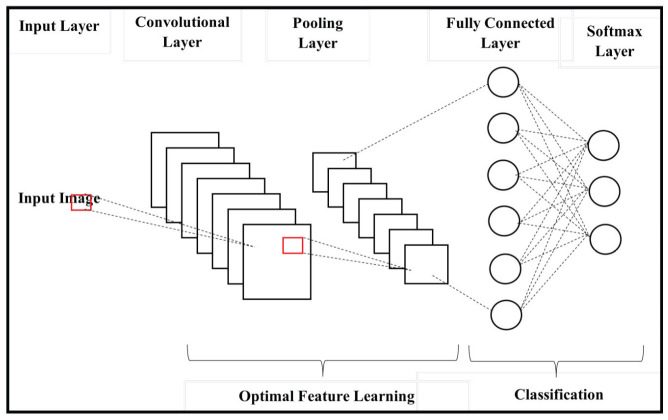
Outline of deep CNN with RELU.

**Figure 4 diagnostics-13-00775-f004:**
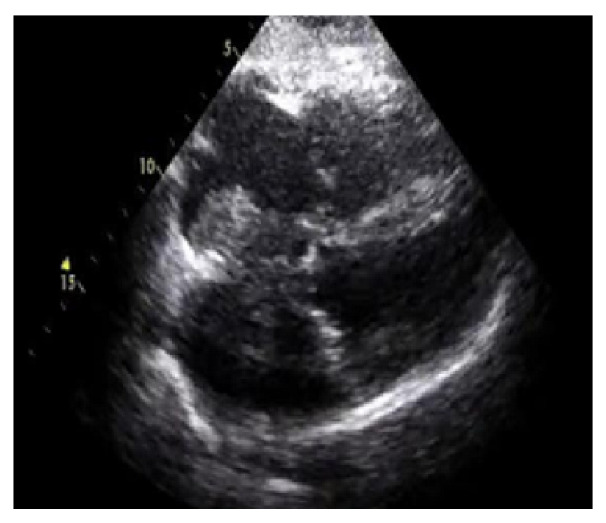
Input image.

**Figure 5 diagnostics-13-00775-f005:**
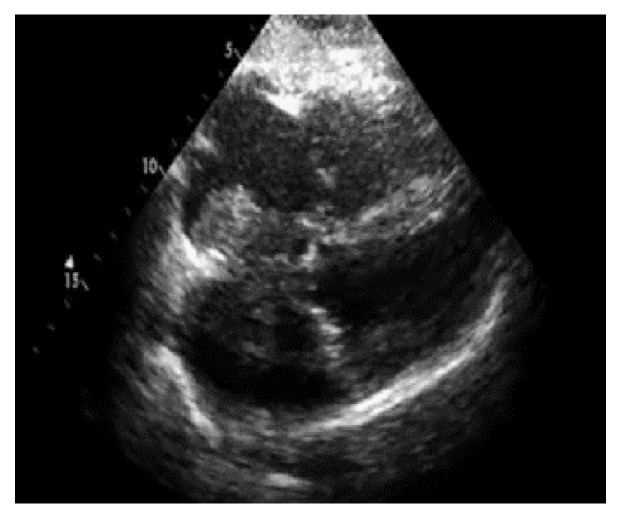
Output of the pre-processed image.

**Figure 6 diagnostics-13-00775-f006:**
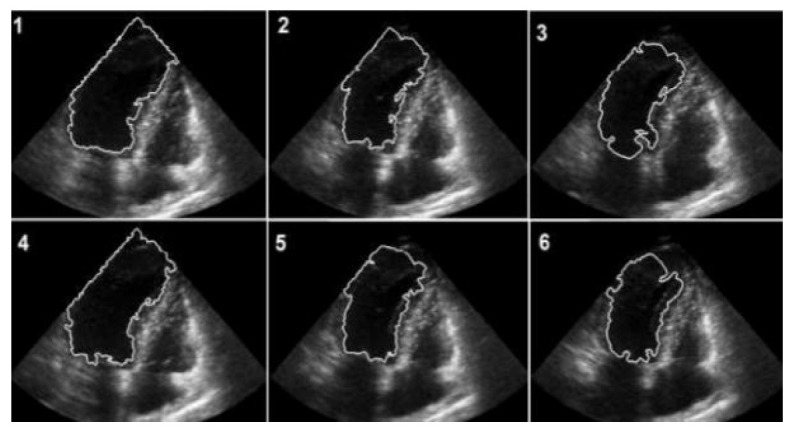
FCM-based segmented images.

**Figure 7 diagnostics-13-00775-f007:**
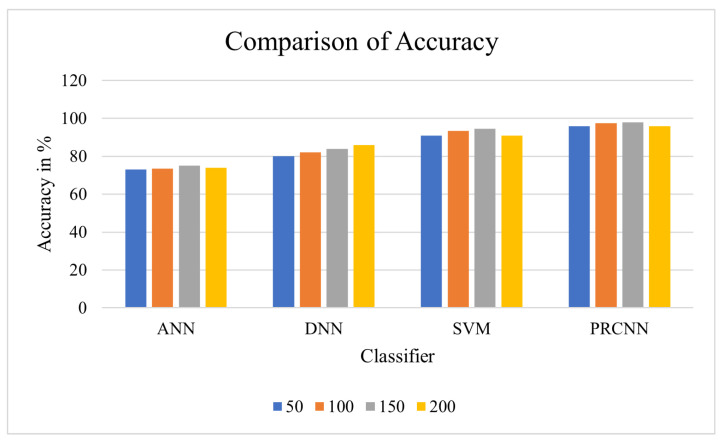
Comparison of accuracy.

**Figure 8 diagnostics-13-00775-f008:**
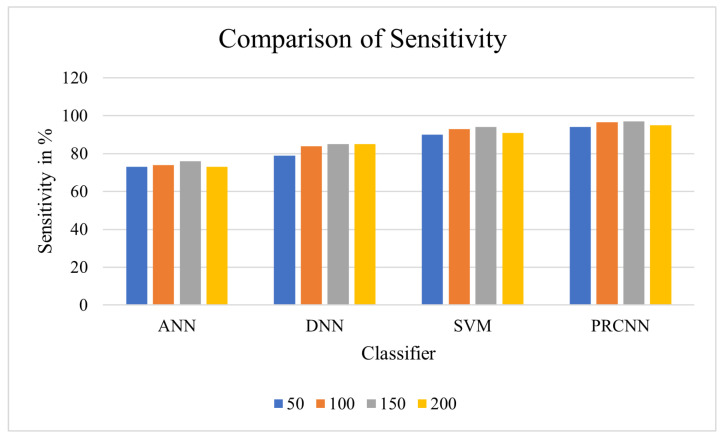
Comparison of sensitivity.

**Figure 9 diagnostics-13-00775-f009:**
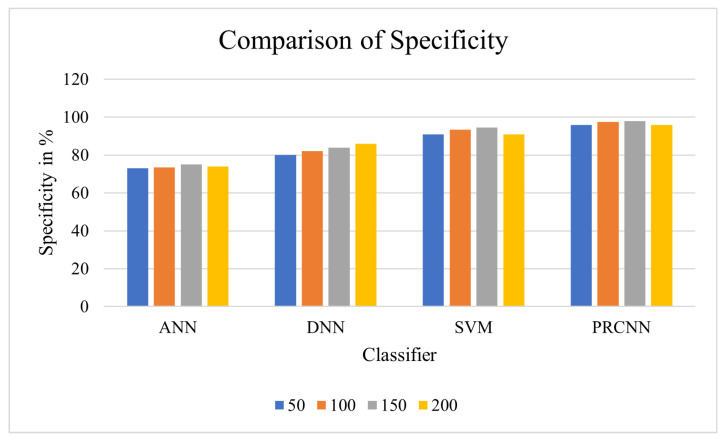
Comparison of specificity.

**Table 1 diagnostics-13-00775-t001:** Cardiovascular diseases: an overview.

Cardiovascular Diseases and Their Types	
Types	Description
Rheumatic heart diseases	Rheumatic fever
Angina	Deficiency in blood supply leads to chest pain and heart muscle weakening
Arrhythmia	Atypical heart rhythm
Coronary artery disease	The arteries are obstructed and the blood supply is stopped
Cardiomyopathy	Disease related to heart muscle
Congenital heart disease	Disfigurements of the heart that are present at birth
Acute coronary syndrome	Supply of blood to the muscles are obstructed suddenly
**Cardiovascular Diseases and their Risk Factors**	
**Risk Factor**	**Study Results**
Gender	When compared to females, males are at high risk
Age	Most old people are affected by the heart disease
Family History	Sometimes the probability of a heart disease diagnosis is hereditary. If any of the individual’s family members have heart disease, then there is a high chance for the occurrence of heart disease.
Poor Diet	Poor dietary habits are necessary for heart disease development
Smoking	Smokers will be highly affected by heart disease
Blood Pressure	Blood pressure thickens blood vessels, and narrows and hardens arteries
Diabetes	Sometimes an outcome of high levels of blood sugar
High blood cholesterol level	Increases plaque formation
Obesity	Being overweight is a reason for heart disease
Stress	Damages the arteries
Physical inactivity	Proper heart functioning is reduced
Poor Hygiene	Increases the chance of heart disease

**Table 3 diagnostics-13-00775-t003:** Denoise level.

Name of Filter	Denoise Level
Median Filter	+8.56 db
Wiener Filter	+9.83 db
Gaussian Filter	+9.39 db
Kuan Filter	+6.98 db

**Table 4 diagnostics-13-00775-t004:** Accuracy comparison analysis between proposed and existing classifiers.

No. of Nodes	ANN	DNN	SVM	PRCNN
50	71	84	85	97
100	74	86	86	98
150	76	87	88	99
200	77	86.5	90	**99.5**

**Table 5 diagnostics-13-00775-t005:** Sensitivity comparison analysis between proposed and existing classifiers.

No. of Nodes	ANN	DNN	SVM	PRCNN
50	73	79	90	94
100	74	84	93	96.5
150	76	85	94	**97**
200	73	85	91	95

**Table 6 diagnostics-13-00775-t006:** Specificity Comparison Analysis between proposed and existing classifiers.

No. of Nodes	ANN	DNN	SVM	PRCNN
50	73	80	91	96
100	73.5	82	93.5	97.5
150	75	84	94.5	**98**
200	74	86	91	96

## Data Availability

This study uses a dataset from the UCI Machine Learning Repository.

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
