# Peer review of "IoT-Enabled Classification of Echocardiogram Images for Cardiovascular Disease Risk Prediction with Pre-Trained Recurrent Convolutional Neural Networks"

_diagnostics, 2023, doi:10.3390/diagnostics13040775_

Round 1
Reviewer 1 Report
This paper uses pre-trained recurrent convolutional neural networks for cardiovascular disease risk prediction based on echocardiogram images. Some comments are as follows:
1. The introduction section is mainly about the research background. However, the research gap in the literature is not identified and the research motivation for this specific study is not justified. For example, what are the existing problems in previous studies and how this study help to solve these problems?
2. The authors should summarize their main contributions and findings in the end of the Introduction section, before the paper structure paragraph.
3. The section name "2. Background Study" is better to be "2. Related Work"
4. Section 2 only presents a list of paper summary reports. It is not well-organized. The authors may want to categorize these studies into different types, e.g., from the methodology perspective. Then identify their advantages and disadvantages.
5. Typos exist and the language quality is below expectation. A full English proofreading is required. For example, the sentence "random forest classifier achieved great accuracy at 98.7 " in line 109, Page 3 is not finished.
6. Section name "3. Classification and Risk Prediction using Deep Learning" is too long. "3. Methodology" should be enough.
7. An overview and a workflow chart should be added in the beginning of Section 3, so that the readers can know the order of different steps.
8. Subfigures in Figure 5. FCM Based Segmented Images are too small.
Author Response
Answers for Comments and Suggestions
Point 1: The introduction section is mainly about the research background. However, the research gap in the literature is not identified and the research motivation for this specific study is not justified. For example, what are the existing problems in previous studies and how this study helps to solve these problems?
Response 1: Thank you for your valuable comment. The research gap and motivation are mentioned in Section 2 (Page no. 6 and Line no.159-166).
Point 2: The authors should summarize their main contributions and findings in the end of the Introduction section, before the paper structure paragraph.
Response 2: The contributions has been summarized in Section 1(Refer the Page no. 3 and Line no.100 to 108 ).
Point 3: The section name "2. Background Study" is better to be "2. Related Work"
Response 3: The correction has been done in the manuscript.
Point 4: Section 2 only presents a list of paper summary reports. It is not well-organized. The authors may want to categorize these studies into different types, e.g., from the methodology perspective. Then identify their advantages and disadvantages.
Response 4: Thank you for the valuable suggestions. We have included the Table 2 and refer Page no.5 that identifies the pros and cons of existing system.
Point 5: Typos exist and the language quality is below expectation. A full English proofreading is required. For example, the sentence "random forest classifier achieved great accuracy at 98.7 " in line 109, Page 3 is not finished.
Response 5: Thank you for the valuable comments. In the revision, overall language is improved throughout all pages in the paper and the line mentioned "random forest classifier achieved great accuracy at 98.7 " is completed and refer Page no.3 in line no.117.
Point 6: Section name "3. Classification and Risk Prediction using Deep Learning" is too long. "3. Methodology" should be enough.
Response 6: Thank you for the valuable comments. The correction has been done in the manuscript.
Point 7: An overview and a workflow chart should be added in the beginning of Section 3, so that the readers can know the order of different steps.
Response 7: Thank you for the valuable comments. The overview and workflow have been added in Section 3 (Refer the Page no. 6).
Point 8: Subfigures in Figure 5. FCM Based Segmented Images are too small.
Response 8: Thank you for the valuable comments. As per the suggested comments, FCM Based Segmented Images are enlarged in Page no.12.
Reviewer 2 Report
- The complexity issues for all the investigated approaches should be included (e.g. the average running times).
- The Pseudo3D GU-Net framework for the segmentation of musculoskeletal imaging applications using multiple deep learning methods should be mentioned.
- English usage should be improved.
- There are a few typos that should be corrected (e.g. "Nave", etc).
- I suggest putting in bold the best results in the relevant tables.
- To facilitate reproducible research, if possible, I suggest that the author release the related source codes on github.com, the website of the authors' research group, or a similar website. This could make a positive impact on the academic community.
Author Response
Answers for Comments and Suggestions
Point 1: The complexity issues for all the investigated approaches should be included (e.g. the average running times).
Response 1: Thank you the valuable comments, complexity issues for all the investigated approaches is done in Section 4 and refer the Page no.14.
Point 2: The Pseudo3D GU-Net framework for the segmentation of musculoskeletal imaging applications using multiple deep learning methods should be mentioned.
Response 2: Thank you the valuable comments, the segmentation algorithm is included in the manuscript. (Algorithm 1).
Point 3: English usage should be improved.
Response 3: Thank you for the suggestions. In the revised manuscript, language usage is enhanced throughout the paper.
Point 4: There are a few typos that should be corrected (e.g. "Nave", etc).
Response 4: Thank you for the suggestions. The typos has been corrected throughout the manuscript.
Point 5: I suggest putting in bold the best results in the relevant tables.
Response 5: Thank you for the suggestions. The changes have been done in the manuscript (Refer the page numbers 12,13 and 14).
Point 6: To facilitate reproducible research, if possible, I suggest that the author release the related source codes on github.com, the website of the authors' research group, or a similar website. This could make a positive impact on the academic community.
Response 6: Thank you for the suggestions. As per my guide’s(supervisor) instruction, this project code has been applied for Indian government copyrights. After the grant approval, this code will be released in github.
Reviewer 3 Report
Paper deals with image segmentation applied to cardiovascular images. The topic is very interesting from both a computer science and medical point of view. The introduction of novel algorithms able to improve medical science. background is well discussed.
Nevertheless I retain that authors must improve the sections: 3.4 since some paragraphs are very trivial and results. I suggest to increase the number of the experiments and also to try the proposed methods in presence of noise.
Author Response
Answers for Comments and Suggestions
Point 1: Paper deals with image segmentation applied to cardiovascular images. The topic is very interesting from both a computer science and medical point of view. The introduction of novel algorithms able to improve medical science. background is well discussed.
Response 1: Thank you for the valuable comments.
Point 2: Nevertheless I retain that authors must improve the sections: 3.4 since some paragraphs are very trivial and results. I suggest to increase the number of the experiments and also to try the proposed methods in presence of noise.
Response 2: Thank you for the valuable suggestions. The experiments are carried out considering the noise also and the summary is presented in Page no.7 and 8.( Refer the line numbers 204 to 218).
Round 2
Reviewer 1 Report
Dear authors,
Thanks for revising and resubmitting the paper. Previous problems are solved and no further comments.
Reviewer 2 Report
The authors have addressed my comments.
Reviewer 3 Report
Authors improved the paper following reviewer suggestions